# SNARE Protein DdVam7 of the Nematode-Trapping Fungus *Drechslerella dactyloides* Regulates Vegetative Growth, Conidiation, and the Predatory Process via Vacuole Assembly

Yue Chen,[a] Jia Liu,[a] Yani Fan,[b,c] Meichun Xiang,[b,c] Seogchan Kang,[d] Dongsheng Wei,[a] Xingzhong Liu[a]

[a]State Key Laboratory of Medicinal Chemical Biology, Key Laboratory of Molecular Microbiology and Technology of the Ministry of Education, Department of Microbiology, College of Life Science, Nankai University, Tianjin, China
[b]State Key Laboratory of Mycology, Institute of Microbiology, Chinese Academy of Sciences, Beijing, China
[c]University of Chinese Academy of Sciences, Beijing, China
[d]Department of Plant Pathology and Environmental Microbiology, The Pennsylvania State University, State College, Pennsylvania, USA

**ABSTRACT** Soluble *N*-ethylmaleimide-sensitive factor attachment protein receptor (SNARE) proteins play conserved roles in membrane fusion events in eukaryotes and have been documented to be involved in fungal growth and pathogenesis. However, little is known about the roles of SNAREs in trap morphogenesis in nematode-trapping fungi (NTF). *Drechslerella dactyloides*, one of the constricting ring-forming NTF, captures free-living nematodes via rapid ring cell inflation. Here, we characterized DdVam7 of *D. dactyloides*, a homolog of the yeast SNARE protein Vam7p. Deletion of *DdVam7* significantly suppressed vegetative growth and conidiation. The mutation significantly impaired trap formation and ring cell inflation, resulting in a markedly decreased nematode-trapping ability. A large vacuole could develop in ring cells within ~2.5 s after instant inflation in *D. dactyloides*. In the Δ*DdVam7* mutant, the vacuoles were small and fragmented in hyphae and uninflated ring cells, and the large vacuole failed to form in inflated ring cells. The localization of DdVam7 in vacuoles suggests its involvement in vacuole fusion. In summary, our results suggest that DdVam7 regulates vegetative growth, conidiation, and the predatory process by mediating vacuole assembly in *D. dactyloides*, and this provides a basis for studying mechanisms of SNAREs in NTF and ring cell rapid inflation.

**IMPORTANCE** *D. dactyloides* is a nematode-trapping fungus that can capture nematodes through a constricting ring, the most sophisticated trapping device. It is amazing that constricting ring cells can inflate to triple their size within seconds to capture a nematode. A large centrally located vacuole is a unique signature associated with inflated ring cells. However, the mechanism underpinning trap morphogenesis, especially vacuole dynamics during ring cell inflation, remains unclear. Here, we documented the dynamics of vacuole assembly during ring cell inflation via time-lapse imaging for the first time. We characterized a SNARE protein in *D. dactyloides* (DdVam7) that was involved in vacuole assembly in hyphae and ring cells and played important roles in vegetative growth, conidiation, trap morphogenesis, and ring cell inflation. Overall, this study expands our understanding of biological functions of the SNARE proteins and vacuole assembly in NTF trap morphogenesis and provides a foundation for further study of ring cell rapid inflation mechanisms.

**KEYWORDS** constricting ring, *Drechslerella dactyloides*, trap formation, ring cell inflation, SNARE, vacuole assembly

Address correspondence to Dongsheng Wei, weidongsheng@nankai.edu.cn, or Xingzhong Liu, liuxz@nankai.edu.cn.

The authors declare no conflict of interest.

Life in eukaryotes depends on communication between membrane-enclosed organelles, which relies on the orderly execution of membrane fusion. The membrane fusion process, carried out by multiprotein complexes, is conserved among eukaryotes and is essential

10.1128/spectrum.01872-22 1

for cell growth and division (1–3). SNARE proteins have been implicated as the conserved and key elements in all intracellular membrane fusion events and have been extensively studied in mammals, plants, and the budding yeast *Saccharomyces cerevisiae* (4, 5). SNAREs are divided into t-SNARE and v-SNARE, which are localized in opposing membranes and form a four-helix bundle that pulls the two membranes tightly together to cause membrane fusion (4). Based on highly conserved residues, SNAREs are further classified as R-SNAREs (arginine-containing SNAREs) or Q-SNAREs (glutamine-containing SNAREs) (6). One characteristic of SNARE proteins is that they contain a domain called the SNARE motif, an evolutionarily conserved stretch of 60 to 70 amino acids in the membrane-proximal regions that are arranged in heptad repeats and form coiled-coil structures (5). Some SNARE proteins have been shown to be involved in the growth and virulence of filamentous fungi. Examples include MoSso1, MoSyn8, MoTlg2, MoSec22, and MoVam7 in the rice blast fungus *Magnaporthe oryzae* (7–11), UmYup1 in the corn smut fungus *Ustilago maydis* (12), GzSyn1/2 and FgVam7 in *Fusarium graminearum* (13, 14), and FoIVam7 in *Fusarium oxysporum* (15). Among them, the proteins orthologous to the *S. cerevisiae* Vam7 protein carry a unique domain called Phox homology (PX) that is responsible for membrane binding. In *S. cerevisiae*, *M. oryzae*, *F. graminearum*, and *F. oxysporum*, the Vam7 has been shown to be involved in vacuole fusion (11, 13, 15, 16), which plays a vital role in fungal growth, differentiation, and pathogenesis (17).

In nature, the ability of animals to hunt prey is well known, and carnivorous plants, such as Venus flytraps, are capable of trapping and digesting small animals, especially insects, through rapid multicellular movements (18). The nematode-trapping fungi (NTF), as representatives of carnivorous fungi, can capture free-living nematodes using different trapping devices, such as adhesive knobs, adhesive nets, nonconstricting rings, and constricting rings (19–21). The trap formation is a morphological feature distinct for NTF. Signal transduction pathways and subcellular compounds involved in trap morphogenesis have been studied in the adhesive net-forming fungus *Arthrobotrys oligospora*, such as the G-protein signaling pathway, the mitogen-activated protein kinase (MAPK) signaling pathway, the cAMP-protein kinase A signaling pathway, the $Ca^{2+}$-related signaling pathway, and the Woronin body, autophagy, and peroxisomes (22). However, the roles of SNAREs in trap morphogenesis in NTF remain poorly understood.

*Drechslerella dactyloides* captures nematodes by a mechanical force created by a constricting ring (CR), which consists of two stalk cells and three ring cells. When a nematode enters the CR lumen, the ring cells inflate to triple their size within seconds to capture the nematode (Fig. 1A; see also Movie S1 in the supplemental material), and then one or more inflated ring cells germinate to penetrate the nematode cuticle layer and digest the body (23–25). A large centrally located vacuole is a unique signature associated with inflated cells (26). We previously showed that deletion of *DdaSte12*, a transcription factor downstream of the MAPK signaling pathway in *D. dactyloides*, caused malformed vacuoles in the ring cells and disabled ring cell inflation (27). However, the mechanism underpinning CR morphogenesis, especially vacuole dynamics during CR formation and ring cell inflation, remains unclear. In this study, we characterized the role of the SNARE protein DdVam7 in vegetative growth, conidiation, and trap morphogenesis, especially ring cell inflation, in *D. dactyloides*.

## RESULTS

**Identification and deletion of *DdVam7* in *D. dactyloides*.** There are approximately 16 putative SNARE proteins encoded by *D. dactyloides* (see Table S1 in the supplemental material). An ortholog of *Vam7* in the *D. dactyloides* genome, *DdVam7*, was identified. Its 1,150-bp open reading frame (ORF) codes for a protein of 361 amino acids, which contain a PHOX homology motif (residues 8 to 113) at the N terminus and a SNARE domain (residues 297 to 340) at the C terminus (Fig. S1A). DdVam7 is highly similar to its ortholog encoded by another CR-forming fungus, *Drechslerella stenobrocha* (77.01%), compared to orthologs encoded by other fungi, such as *Fusarium oxysporum* f. sp. *lycopersici* (39.06%), *Aspergillus oryzae* (42.66%), and *Saccharomyces cerevisiae* (16.77%) (Fig. S1B).

After inoculation with *Caenorhabditis elegans*, a large number of CRs formed at 8 to 12 h. These CRs required several hours of maturation to be capable of inflation. At 24 h, the CRs

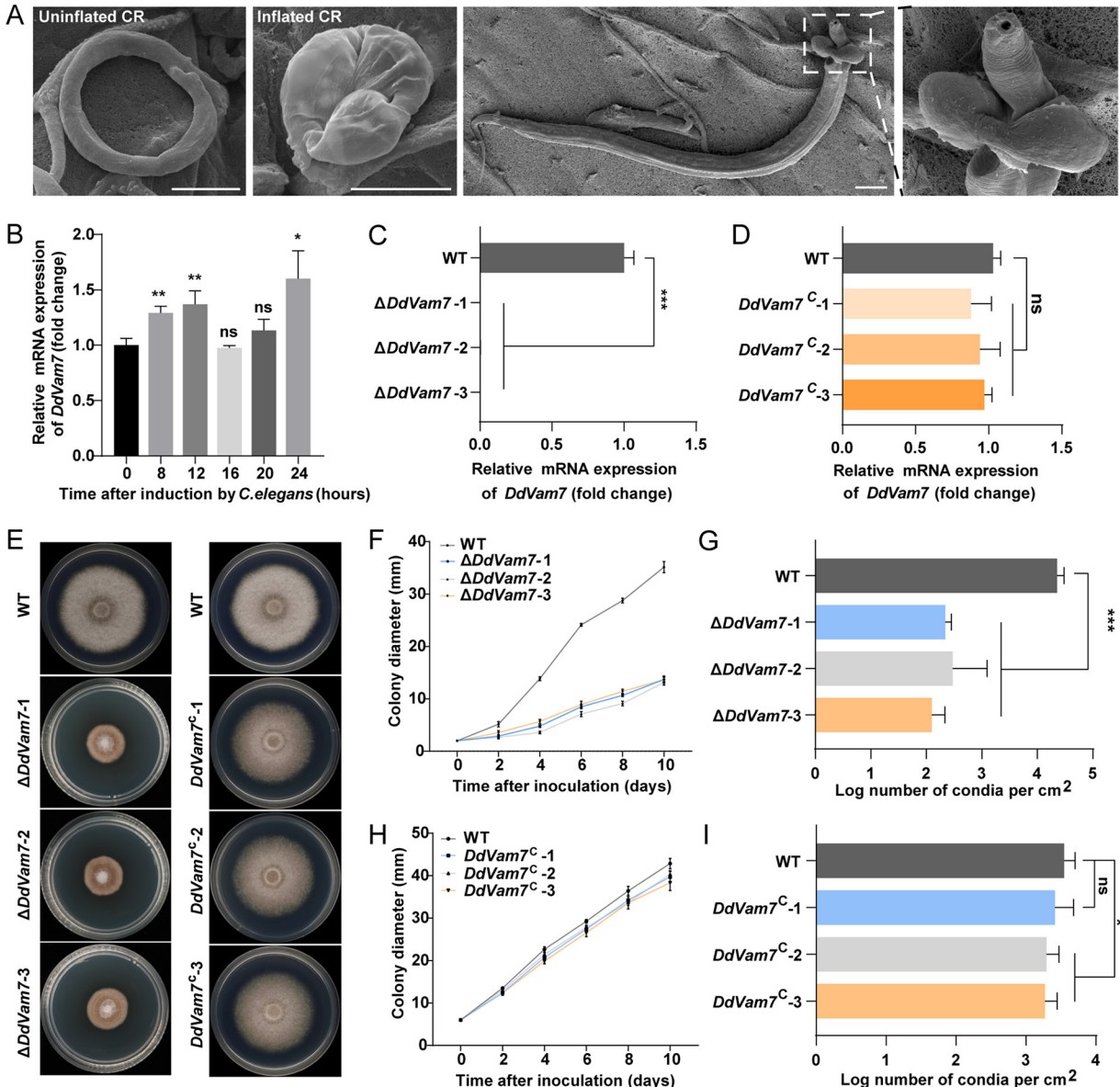

**FIG 1** Expression of *DdVam7*, colony growth, and conidiation of the wild-type (WT), Δ*DdVam7*, and *DdVam7*$^C$ strains. (A) The uninflated CR, inflated CR, and a nematode captured by a CR in *D. dactyloides*. Scale bars, 10 μm. (B) Expression of *DdVam7* at 0, 8, 12, 16, 20, and 24 h after inoculating *C. elegans*. (C) *DdVam7* expression in WT and three Δ*DdVam7* mutants. (D) *DdVam7* expression in WT and three *DdVam7*$^C$ strains. For panels B, C, and D, ***, $P < 0.001$; **, $P < 0.01$; *, $P < 0.05$; ns, not significant ($n = 4$). (E) Colony morphology of WT, Δ*DdVam7*, and *DdVam7*$^C$ strains cultured on PDA plates for 10 days. Data shown are representative of experiments performed in duplicate. (F) Colony diameters of WT and three Δ*DdVam7* mutants. (G) Conidiation by WT and three Δ*DdVam7* mutants on PDA plates after incubation for 10 days. (H) Colony diameters of WT and three *DdVam7*$^C$ strains. (I) Conidiation of WT and *DdVam7*$^C$ strains cultured on PDA plates for 10 days. In panels F to I, ***, $P < 0.001$; *, $P < 0.05$; ns, not significant ($n = 5$).

were able to inflate and capture nematodes (Fig. S2). Expression of *DdVam7* was significantly upregulated at 8, 12, and 24 h after inoculating *C. elegans* (Fig. 1B), supporting the involvement of DdVam7 in CR formation and inflation. We deleted *DdVam7* in *D. dactyloides* via homologous recombination. Three independently isolated knockout mutants (Δ*DdVam7*-1, Δ*DdVam7*-2, and Δ*DdVam7*-3), verified by PCR and reverse transcription-PCR (RT-PCR), were selected from 77 transformants (Fig. S3A; Fig. 1C). For the complementation, the mutant strain was transformed with a plasmid carrying the full-length *DdVam7* gene under the control of the native promoter. Three complementary strains (*DdVam7*$^C$-1, *DdVam7*$^C$-2, and *DdVam7*$^C$-3) that restored *DdVam7* expression were selected from 48 transformants (Fig. S3B; Fig. 1D).

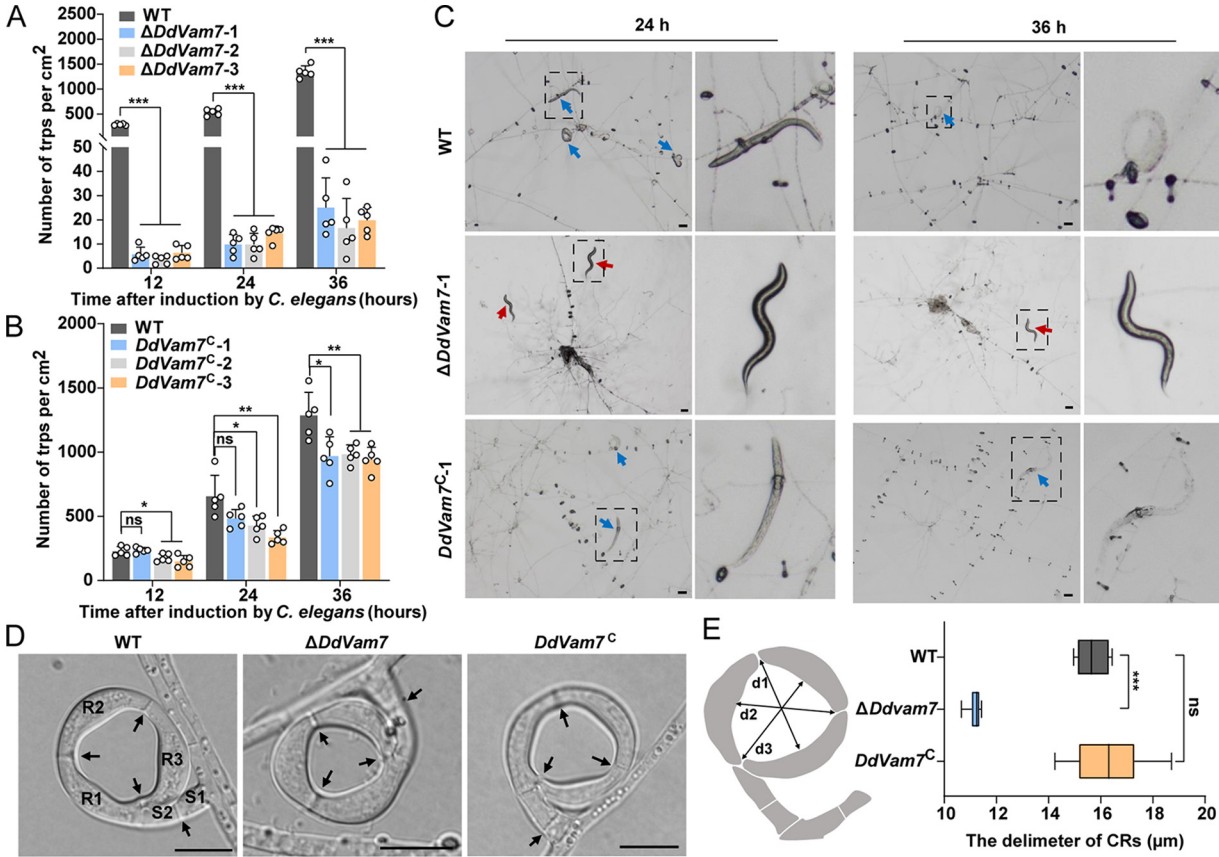

**FIG 2** CR formation by wild-type (WT), Δ*DdVam7*, and *DdVam7*[C] strains. (A) Bar chart showing the numbers of CRs formed at 12, 24, and 36 h after inoculating *C. elegans* in WT and Δ*DdVam7*. (B) Bar chart showing the CR numbers at 12, 24, and 36 h after inoculation of *C. elegans* in WT and *DdVam7*[C]. In panels A and B, ***, $P < 0.001$; **, $P < 0.01$; *, $P < 0.05$; ns, not significant ($n = 5$). (C) Light micrographs showing CR formation capacity. Red arrows indicate free nematodes, and blue arrows denote captured nematodes. The right panel is a close-up view of the area indicated by the dotted black box in the left panel. Shown are representatives of at least five images. Scale bars, 50 μm. (D) Respective micrographs showing the morphological characteristics of CRs. R1, R2, and R3, ring cells; S2 and S2, stalk cells. Black arrows denote the septa in CRs. Scale bars, 10 μm. (E) Box plot showing the diameter distribution of CRs. The diameter of the CRs was averaged at division 1 (d1), d2, and d3, as shown in the left schematic. ***, $P < 0.001$; ns, not significant ($n = 15$).

**DdVam7 is involved in vegetative growth and conidiation.** Mycelial growth and conidiation were significantly reduced when *DdVam7* was deleted. Compared with the wild-type and complemented strains, Δ*DdVam7* colonies were compact and button-like with sparse aerial hyphae (Fig. 1E). After 10 days of growth on potato dextrose agar (PDA), the average colony diameter of Δ*DdVam7* was ~13 mm, while that of the wild type was ~35 mm (Fig. 1F). Conidiation by Δ*DdVam7* was only ~40% of that with the wild type after 10-day incubation (Fig. 1G). Complementation of Δ*DdVam7* restored colony growth and conidiation (Fig. 1H and I). Sensitivities to osmotic stress and wall stressors were also evaluated for the wild type and Δ*DdVam7* mutant. Relative growth inhibition values of the wild type and Δ*DdVam7* in the presence of 0.5 M NaCl were 66.1% and 42.3%, while 48.7% and 20% inhibition was noted under 1 M sorbitol, indicating that Δ*DdVam7* was less sensitive to osmotic stresses (Fig. S4A and B). Compared to the wild type, Δ*DdVam7* exhibited heightened sensitivity during growth on PDA plates containing 0.05% Congo red, but the deletion mutant showed lower relative growth inhibition under SDS treatment (Fig. S4A and B). We did not observe notable differences in the cell wall between the wild type and Δ*DdVam7* when they were stained using calcofluor white and concanavalin A (Fig. S4C).

**Deletion of *DdVam7* impairs CR formation and ring cell inflation.** Deletion of *DdVam7* significantly reduced trap formation. The trap numbers of Δ*DdVam7* and the wild type were ~11 versus ~537/cm² at 24 h after inoculating *C. elegans* and ~20 versus ~1,343/cm² at 36 h (Fig. 2A and C). Complementation of *DdVam7* mostly restored the ability of CR formation, with the trap numbers of ~455 versus ~626/cm² (24 h after inoculating *C.*

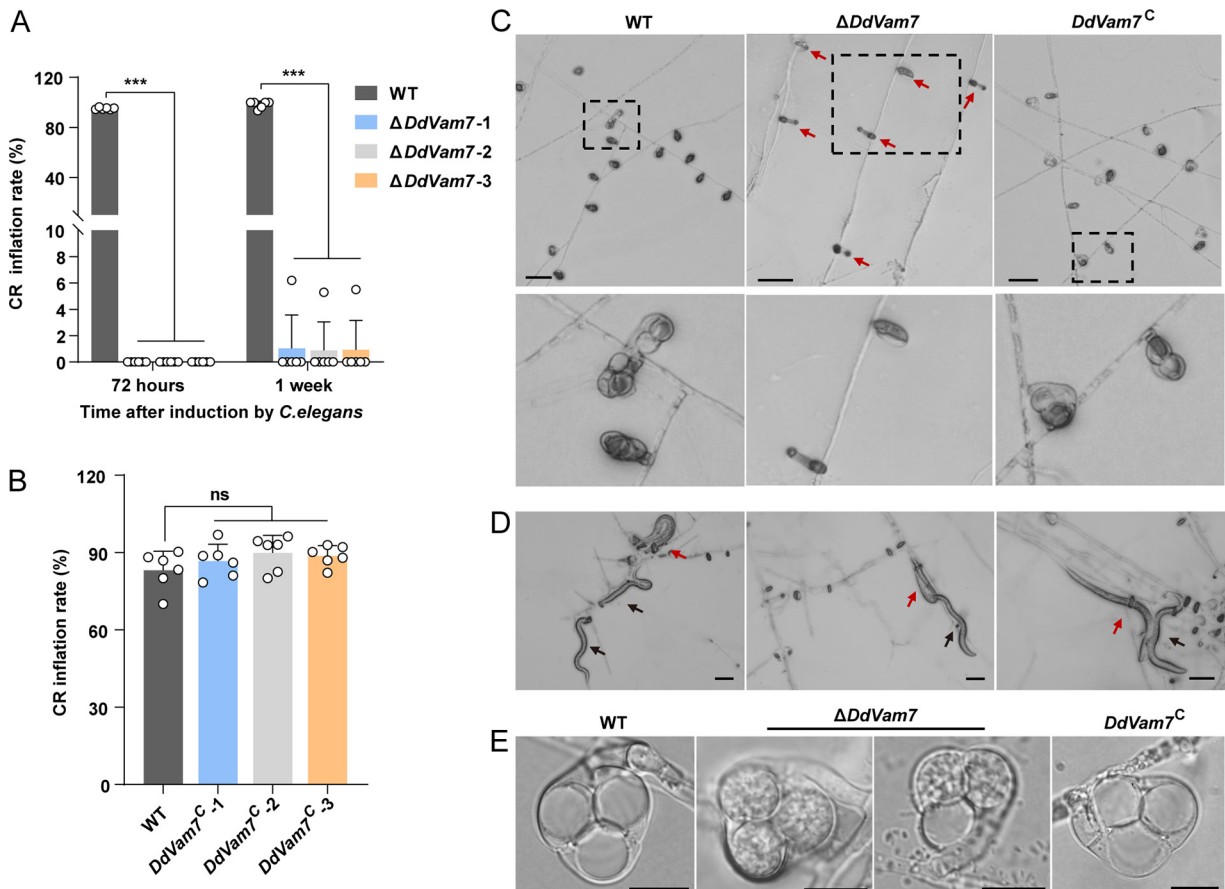

**FIG 3** Requirement of DdVam7 for ring cell inflation. (A) Bar chart showing the percentages of inflated CRs at 72 h and 1 week after inoculating *C. elegans* into cultures of the wild type (WT) and Δ*DdVam7*. (B) Bar chart showing the percentages of inflated CRs at 72 h after inoculating *C. elegans* into cultures of the WT and *DdVam7*<sup>C</sup>. In panels A and B, ***, $P < 0.001$; ns, not significant ($n = 6$). (C) Light micrographs showing ring cell inflation at 72 h after inoculating *C. elegans*. Red arrows indicate uninflated CRs. The lower panel shows close-up views of the areas indicated by the dotted black box in the upper panel. Shown are representatives of at least five images. Scale bars, 50 μm. (D) Light micrographs showing the nematodes captured by Δ*DdVam7* CRs. Red arrows indicate captured nematodes, and black arrows denote free nematodes. Scale bar, 50 μm. (E) Respective micrographs showing inflated CRs of WT, Δ*DdVam7*, and *DdVam7*<sup>C</sup>. Scale bar, 10 μm.

elegans) and ~963 versus ~1,296/cm² (36 h after inoculating *C. elegans*) for *DdVam7*<sup>C</sup> and wild type, respectively (Fig. 2B and C). In addition to decreased CR numbers, the diameters of Δ*DdVam7* CRs were significantly smaller than those of the wild type (Fig. 2D and E). However, the number of septa and cells of CRs in the wild type and Δ*DdVam7* showed no differences (Fig. 2D). Although the CRs of Δ*DdVam7* mutants could attract nematodes to pass through their lumen, they failed to inflate to capture them at 36 h after inoculating *C. elegans* (Fig. 2C; Movies S1 to S3). CRs of Δ*DdVam7* also failed to inflate even by heat stimulation at 72 h after inoculating *C. elegans*, whereas 83% CRs of the wild type inflated (Fig. 3A and C). However, a few CRs of Δ*DdVam7* could inflate in response to nematodes or hot water stimulation after 1 week of induction (Fig. 3A, D, and E). The inflation defect could be restored by complementation (Fig. 3B and C). Consistently, the trap formation and ring cell inflation were also significantly impaired in the Δ*DdVam7* mutant, which showed growth comparable to the wild type. The 25-day-old culture of Δ*DdVam7* reached the colony diameter of ~22 mm, which was comparable to that of the 3-day-old wild-type culture (Fig. S5A and B). The numbers of traps formed by Δ*DdVam7* were only 8.9% (24 h after inoculating *C. elegans*) and 9.5% (36 h after inoculating *C. elegans*) of those formed by wild type (Fig. S5C and D). CRs of Δ*DdVam7* failed to inflate by heat stimulation at 72 h after inoculating *C. elegans*, whereas 98% CRs of the wild type inflated. A few CRs of Δ*DdVam7* could inflate in response to nematodes or hot water stimulation after 1 week of induction (Fig. S5E and F).

**DdVam7 regulates vacuole assembly in *D. dactyloides*.** To investigate the intracellular localization of DdVam7, a strain with a red fluorescent protein (RFP) tag at the N terminus of DdVam7 (RFP-DdVam7) in the Δ*DdVam7* mutant genetic background was constructed (Fig. S6A). In Δ*DdVam7/RFR-DdVam7*, vegetative growth and CR formation were largely restored (Fig. S6B and E), and its ring cell inflation was completely restored (Fig. S6F and G). DdVam7 fused with RFP accumulated in punctate structures of the apical regions of hyphae and in large organelles of the basal regions of hyphae (Fig. 4A). Colocalization of RFP signal with that of 7-amino-4-chloromethylcoumarin (CMAC), a fluorescent dye that stains the interior of vacuoles, indicated that DdVam7 was located in vacuoles (Fig. 4A). The vacuoles of Δ*DdVam7* were small and fragmented, whereas the vacuoles of the wild type and *DdVam7*^C were regular and large (Fig. 4B and C). Detailed observation showed that small-vacuole fusion had been initiated but failed to be completed in Δ*DdVam7* (Fig. 4C), indicating the involvement of DdVam7 in vacuole fusion in *D. dactyloides*.

A notable morphological trait of CR is a large vacuole that develops in ring cells within ~2.5 s of inflation (Fig. 4D; Movie S4), whereas the large vacuole fails to form in incompletely inflated ring cells (Fig. 4E; Movie S5). Transmission electron microscopy (TEM) analysis showed that the vacuoles in uninflated ring cells were small and regular, numerous irregular spheroids were observed in partially inflated ring cells, and large vacuoles filled the inflated ring cells (Fig. 4F). The vacuoles in uninflated ring cells of Δ*DdVam7* were malformed, and no large and round vacuoles were formed in inflated ring cells (Fig. 4G, Fig. 3E), while each inflated ring cell of the wild type and the complemented strain was almost filled with a large vacuole (Fig. 4G).

**PX and SNARE domains are important for function of DdVam7.** To explore the functions of the PX and SNARE domains, *DdVam7*^ΔPX and *DdVam7*^ΔSNARE deletion domain constructs were generated and transformed into Δ*DdVam7* (Fig. 5A; Fig. S7). Colonies of the domain deletion strains exhibited moderate growth rates between wild type and Δ*DdVam7*. After 10 days of growth on PDA, the average colony diameters of wild type and Δ*DdVam7* were ~35 mm and ~13 mm, respectively, while that of Δ*DdVam7/DdVam7*^ΔPX and Δ*DdVam7/DdVam7*^ΔSNARE were ~22 mm and ~18 mm, respectively (Fig. 5B and C). The PX and SNARE domain-truncated strains displayed an impaired CR formation capacity similar to that of Δ*DdVam7* at 24 h after inoculating *C. elegans* (Fig. 5D and E). After 36 h, the domain deletion strains could produce more traps than Δ*DdVam7*. The trap numbers of Δ*DdVam7/DdVam7*^ΔPX, Δ*DdVam7/DdVam7*^ΔSNARE, and Δ*DdVam7* were ~55, ~44, and ~24/cm², respectively (Fig. 5D and E). Compared with wild type and *DdVam7*^C, the CRs of the SNARE domain deletion strain failed to inflate to capture nematodes at 36 h after inoculating *C. elegans* and also failed to inflate even by heat stimulation at 72 h after inoculating *C. elegans* (Fig. 5E to G); these results were similar to those with Δ*DdVam7*. In contrast, the PX domain deletion strain could inflate to capture nematodes similar to the wild type and *DdVam7*^C at 36 h after inoculating *C. elegans* (Fig. 5E). At 72 h after inoculating *C. elegans*, 97% of CRs of the PX domain deletion strain could inflate by heat stimulation (Fig. 5F and G). These findings indicated that both the SNARE and PX domains of DdVam7 participate in growth and trap formation, but only the SNARE domain plays an important role in ring cell inflation.

## DISCUSSION

Trapping devices are the distinct morphological phenotype and ecological trait for carnivorous fungi (28). Many attempts have been conducted to determine key genes and pathways specifically associated with trap formation and the carnivorous lifestyle. A number of genes involved in multiple signal transduction pathways, nitrate assimilation, intercellular communication, peroxisomes, autophagy, and pH sensing have been investigated in the model trapping fungus *A. oligospora*. Most of the genes involved in trap morphogenesis also participate in vegetative growth, asexual and sexual reproduction, and virulence (22). Examples include, *Ste12*, *Fus3*, and *Bck1* in MAPK pathway (29, 30), *AoAtg8*, *AoAtg5*, and *AoAtg1* involved in autophagy (31–33), *AoPEX1* and *AoPEX6* involved in peroxisome biogenesis (34), and E3-ligase encoding gene *AoUBR1* (35). Some genes are specifically associated with trap formation and trapping efficiency, and their disruption can totally lead to an inability for trap formation, including *AoSlt2* (36), *AoRab-7A* (37), and *AoFIG_2* (38). However, the role of

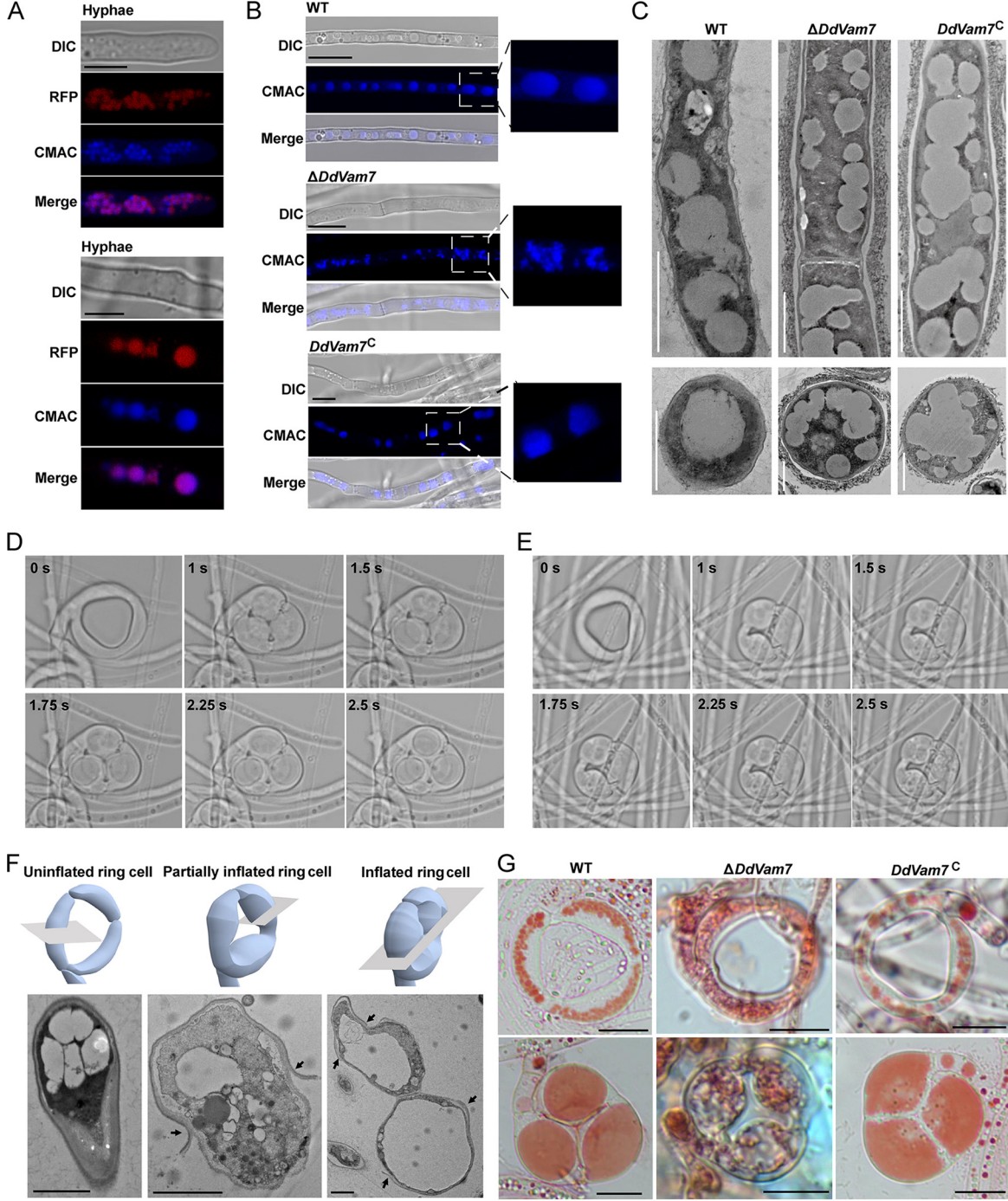

**FIG 4** Involvement of DdVam7 in vacuole assembly. (A) Micrographs showing that DdVam7 localizes in vacuoles in hyphae. Scale bar, 5 μm. (B) Hyphae of the wild type (WT), ΔDdVam7, and DdVam7^C strains stained with CMAC. Scale bar, 5 μm. The images shown here are representatives from experiments performed in duplicate. (C) Transmission electron micrographs showing vacuoles in WT, ΔDdVam7, and DdVam7^C. Shown are representatives of at least 5 images. Scale bar, 2 μm (upper panel) or 1 μm (lower panel). (D and E) Time-lapse images showing the inflation process of a completely inflated CR (D) and a partially inflated CR (E) in the wild type. (F) Transmission electron micrographs showing the vacuoles in uninflated, partially inflated, and completely inflated ring cells in the wild type. Black arrows indicate the ruptured outer cell wall. Shown are representatives of at least 5 images. Scale bar, 2 μm. (G) Vacuoles in ring cells stained with neutral red before (BH) and after (H) heat stimulation in the wild type, ΔDdVam7, and DdVam7^C strains. Shown are representatives from experiments performed in duplicate. Scale bar, 10 μm.

SNARE family proteins in fungal growth, trap morphogenesis, and pathogenicity remain poorly understood in NTF. In this study, an ortholog of SNARE protein Vam7 was characterized in NTF for the first time. Multiphenotype analysis results showed that DdVam7 plays a crucial role in diverse phenotypic traits in constricting the ring-forming fungus *D. dactyloides*.

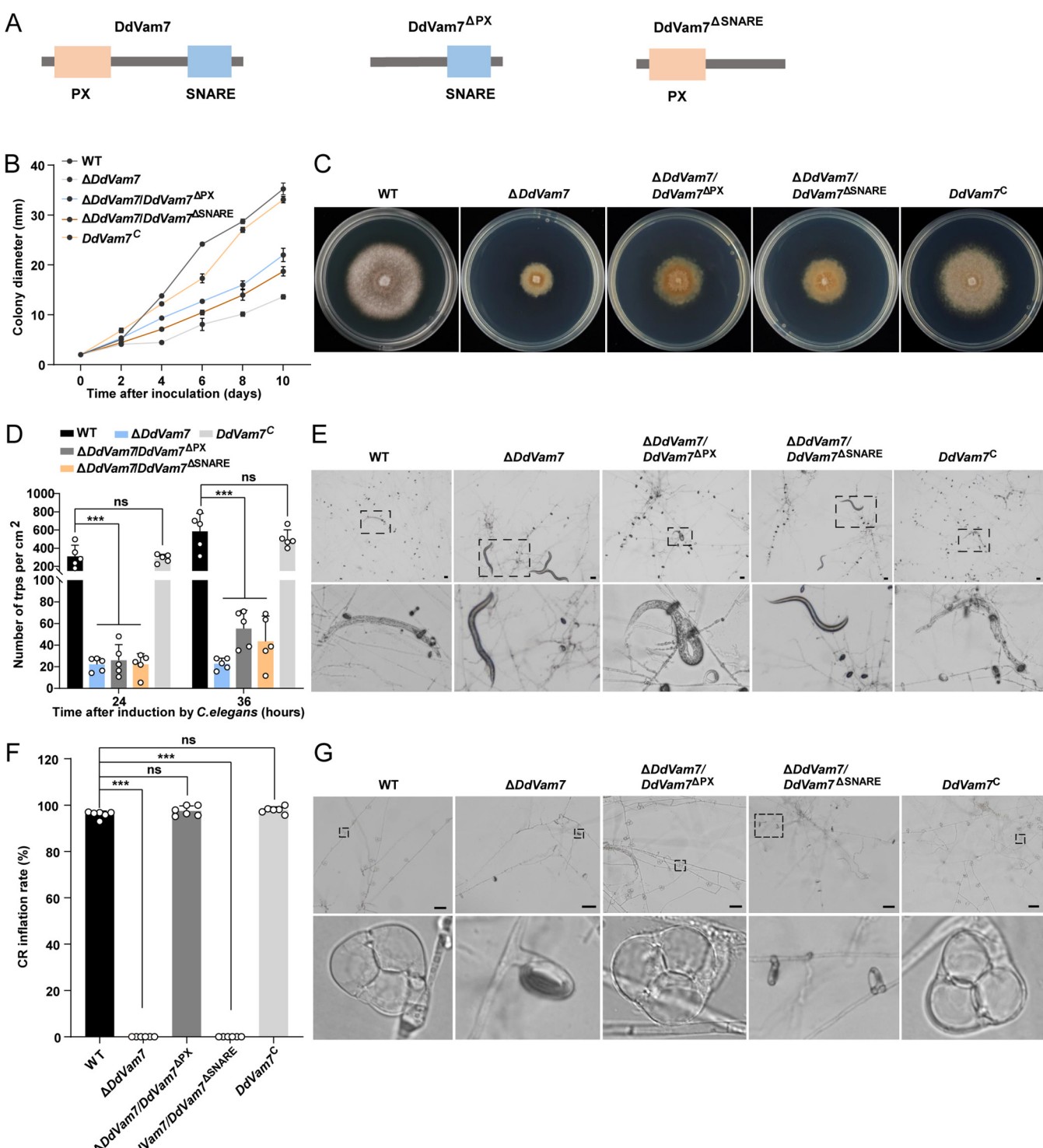

**FIG 5** The PX and SNARE domains are important for the function of DdVam7. (A) Schematic showing domain deletions of DdVam7. (B) Colony diameters of wild-type (WT), Δ*DdVam7*, *DdVam7*<sup>C</sup>, and domain deletion strains. (C) Colony morphology of WT, Δ*DdVam7*, *DdVam7*<sup>C</sup>, and domain deletion strains cultured on PDA plates for 10 days. (D) Bar chart showing the numbers of CRs formed at 24 and 36 h after inoculating *C. elegans*. ***, $P < 0.001$; ns, not significant ($n = 5$). (E) Light micrographs showing CR formation capacity at 36 h after inoculating *C. elegans*. The lower panel is a close-up view of the area indicated by the dotted black box in the upper panel. Shown are representatives of at least five images. Scale bars, 50 $\mu$m. (F) Bar chart showing the percentages of inflated CRs at 72 h after inoculating *C. elegans*. ***, $P < 0.001$; ns, not significant ($n = 6$). (G) Light micrographs showing ring cell inflation at 72 h after inoculating *C. elegans*. The lower panel shows close-up views of the areas indicated by the dotted black box in the upper panel. Shown are representatives of at least five images. Scale bars, 50 $\mu$m.

Fungal vacuoles are dynamic, as they undergo a continuous balance of fission and fusion to allow changes in shape, size, and number and are vital for fungal morphogenesis (39, 40). During hyphal growth, the cytoplasm-filled tip cell extends and leaves behind an empty-looking highly vacuolated cell compartment (41). SNARE family proteins are highly conserved and are involved in membrane fusion events. Among them, Vam7 mediates vacuole fusion to play a crucial role in diverse phenotypic traits of fungi (42). MoVam7 in *M. oryzae* is involved in maintaining the shapes of vacuoles and contributes to the regulation of cell wall integrity, endocytosis, conidiation, and pathogenicity (11). FgVam7 participates in vacuolar maintenance and endocytosis and consequently regulates vegetative growth, conidiation and conidial germination, sexual reproduction, and virulence in *F. graminearum* (13). FolVam7 of *F. oxysporum* is involved in intracellular trafficking, thereby regulating vegetative growth, asexual development, conidial morphology, and plant infection (15). In *Colletotrichum fructicola*, CfVam7 participates in vacuole fusion, appressorium formation, and pathogenicity (43). Here, similar results were found in *D. dactyloides*, as DdVam7 was involved in vacuole fusion. Deletion of *DdVam7* significantly impaired hyphal growth, conidiation trap formation, and ring cell inflation. In *F. oxysporum* and *C. fructicola*, Vam7 has been reported to form ring-shaped structures colocalizing with the vacuole membrane, indicating localization at the vacuole membrane (15, 43). Here, we showed that DdVam7 accumulated in punctate structures of apical regions of hyphae and in large organelles of the basal regions of hyphae (Fig. 4A). Similar results have been reported in *F. graminearum*; FgVam7 is observed mainly in the vacuoles of vegetative hyphae (13). However, these results do not exclude the possibility that FgVam7 and DdVam7 are also located on the vacuolar membrane and may be involved in vacuole fusion. It is possible that the SNARE protein Vam7 may have a different regulatory mechanism in different fungi, playing distinct roles in different stages of exocytosis or endocytosis, such as membrane fusion or vacuole assembly and sorting. The Vam7 protein consists of a SNARE and a PX domain, which are essential for biological functions of Vam7. The PX motif and SNARE domain are required for the pathogenicity of *F. oxysporum* (15). In *C. fruticola*, the SNARE domain is described as essential for pathogenicity, while the PX motif is indicated to play a minor role in virulence (43). Our results show that the SNARE domain but not the PX domain is necessary for ring cell inflation. The SNARE domain is crucial for the function of Vam7, and the PX domain seems to have different regulation mechanisms in the functions of the protein in different fungi.

The trapping mechanisms of carnivorous fungi can be divided into two types: (i) adhesive traps, including adhesive nets, adhesive knobs, and adhesive columns that capture nematodes through the adhesive layer covering the device surface, and (ii) mechanical traps (CRs), which capture the nematodes through mechanical force via rapid ring cell inflation (20, 25, 28). However, the mechanism of ring cell inflation remains unclear. Here, our results clearly showed vacuole assembly in ring cells after inflation, and it took ~2.5 s. Very little is known about the mechanistic background of the formation of the large vacuole during rapid ring cell inflation. Much of our understanding of the role of vacuole assembly in rapid and large cell volume changes has been derived from research on guard cells (44, 45). In *Vicia faba*, guard cells can also undergo large changes in cell volume in a short period of time during stomatal movement. It has been suggested that the small vacuoles can fuse with each other or with the bigger vacuoles to generate large vacuoles, and vacuole volume change makes a major contribution to the change of guard cell volume (45). CR inflation is completed in a very short time, which has resulted in technical difficulties in determining whether vacuole fusion is involved in CR inflation. What can be observed is that vacuoles reassemble during inflation, forming large and round vacuoles that fill almost the entire ring cell, possibly generating enough mechanical force to tightly trap the nematode. With the mutant ΔDdVam7, the vacuoles of uninflated CRs were malformed, and no large and round vacuoles formed in inflated ring cells. Vacuoles are crucial for nutrient transport in fungi, and abnormal vacuoles may interfere with CR maturation, resulting in only a few CRs that can inflate at 1 week after inoculating *C. elegans*. However, lack of large vacuoles in these inflated CRs suggests a potential role of Vam7 in the formation of large vacuoles. The mechanisms underlying the formation of large vacuoles and the molecules directly involved remain to be further explored.

Studies on *Candida albicans* have shown that cells grow by expanding the vacuolar space rather than synthesizing new cytoplasm and organelles under severely nutrient-limited conditions (17, 46). Compared to synthesis of new cytoplasm, vacuole biogenesis is energetically less costly; therefore, hypha space-filling by vacuoles is an important part of fungal morphogenesis (17). For the rapid morphogenesis of ring cells in *D. dactyloides*, the large-vacuole biogenesis appears to drive the instantaneous increase in cell volume, while the mechanism of rapid vacuole dynamics remains to be investigated.

In summary, DdVam7 participates in vacuole fusion and is required for vegetative growth, conidiation, CR formation, and ring cell inflation in *D. dactyloides*. This study expands our understanding of biological functions of the SNARE proteins and vacuole assembly in NTF and provides a valuable framework for future studies focusing on membrane fusion in trap morphogenesis and ring cell rapid inflation.

## MATERIALS AND METHODS

**Strains and culture conditions.** The wild type and its mutants were maintained on PDA (BD, New York, USA) plates at 25°C. For conidiation, strains were cultured on corn meal agar (CMA; BD) plates at 25°C. Water agar (WA; BD) plates were used to induce trap formation with *C. elegans*. For stress assays, strains were cultured on PDA with different concentrations of chemical stressors and incubated at 25°C. *C. elegans* was grown on nematode growth medium (NGM) plates seeded with *Escherichia coli* strain OP50 at 23°C.

**Identification and sequence analysis of DdVam7.** The SNARE protein DdVam7 in *D. dactyloides* was identified from NCBI based on the corresponding amino acid sequences in the model fungus *S. cerevisiae* (Vam7; accession number NP_011303.1). Prediction of conserved functional domains was performed using InterProScan (http://www.ebi.ac.uk/interpro/) and PFAM (http://pfam.xfam.org/) (47). Clustalomega (https://www.ebi.ac.uk/Tools/msa/clustalo/) was used to characterize protein families and detect homology (48).

**Gene deletion, complementation, and domain deletion.** Fungal mycelia after culturing in potato dextrose broth (PDB; BD) for 7 days were used to extract genomic DNA. Gene deletion via homologous recombination was performed using *Agrobacterium*-mediated transformation (27). Two fragments (2,846 bp and 2,875 bp) corresponding to the 5′ and 3′ regions of the *DdVam7* gene were amplified by PCR with the primer sets Vam7-5F/Vam7-5R and Vam7-3F/Vam7-3R from *D. dactyloides* genomic DNA, respectively. These fragments were inserted into both sides of the *hph* resistance gene cassette in pAg1-H3-HygR to produce a gene knockout construct. Hygromycin B-resistant colonies were transferred to PDA medium supplemented with 100 $\mu$g/mL hygromycin B. Three mutants were isolated, and one mutant strain with stable phenotype was selected for subsequent experiments. For complementation, a 5,167-bp PCR product containing the full-length *DdVam7* coding region and the 2,008-bp upstream and 2,009-bp downstream regions was amplified using primers Vam7-CF/Vam7-CR and subcloned into pAg1-H3-NeoR, a modified pAg1-H3-HygR vector containing the NeoR gene, a gene conferring Geneticin resistance (G418), for transforming the mutant strain (Δ*DdVam7*-1). Transformants were selected using PDA medium containing 200 $\mu$g/mL Geneticin. Putative transformants were verified by PCR analysis, and the positive transformants were further confirmed by RT-PCR. The primers used are shown in Fig. S3 and Table S2 in the supplemental material. All mutants were serially transferred for at least 5 generations on PDA containing 100 $\mu$g/mL hygromycin B or 200 $\mu$g/mL Geneticin to purify them. For domain deletion, the DdVam7$^{\Delta PX}$ coding region (744 bp) and DdVam7$^{\Delta SNARE}$ coding region (888 bp) were amplified by primers SNA-F/SNA-R and PX-F/PX-R and subcloned into pAg1-H3-NeoR for the transforming mutant strain (Δ*DdVam7*-1). Transformants were selected using PDA medium containing 200 $\mu$g/mL Geneticin. Putative transformants were verified by PCR analysis. The primers used are shown in Fig. S6A and Table S2.

**Subcellular localization of DdVam7.** To determine cellular localization of DdVam7 in *D. dactyloides*, a transformation cassette containing the entire DdVam7 coding sequence fused with RFP was constructed. A 5,167-bp fragment that covered the *DdVam7* ORF (1,150 bp) and the putative promoter (2,008 bp) and terminator (2,009 bp) regions was amplified using primers Vam7-CF and Vam7-CR. The RFP gene was fused to the N terminus of the *DdVam7* ORF. The resulting construct was cloned into pAg1-H3-NeoR by a modified QuikChange method (49). The resulting plasmid was transferred into the Δ*DdVam7* mutant. Transformants were selected using 200 $\mu$g/mL Geneticin. The primers used are shown in Table S2.

**RNA preparation and RT-PCR analysis.** Conidia of *D. dactyloides* were plated on WA plates covered with cellophane membrane and incubated for 5 days. After treating the resulting cultures with *C. elegans* from 0 to 24 h, they were collected for RNA extraction. Total RNA was extracted using TRIzol reagent (Invitrogen, Carlsbad, USA). The RNA samples were reverse transcribed using All-in-One First-Strand cDNA Synthesis SuperMix (TransGen, Beijing, China) to produce complementary DNA. Real-time PCRs were performed using SYBR PCR mix (TransGen, Beijing, China), and the data were normalized to $\beta$-tubulin expression. These experiments were performed in triplicate. The expression fold changes were calculated using the $2^{-\Delta\Delta CT}$ method.

**Comparison of mycelial growth and conidiation.** Mycelia plugs of the wild-type and mutant strains (6 mm in diameter) were inoculated on PDA medium at 25°C for 10 days to compare their growth. The diameter of each colony was measured every 24 h. The same mycelial plugs were inoculated on CMA plates at 25°C for 10 days before collecting conidia using 5 mL water, and the number of conidia per square centimeter of each colony was calculated. To examine how *DdVam7* deletion affected resistance to environmental stresses, mycelia plugs were inoculated on PDA medium containing 0.5 M NaCl, 1 M sorbitol, 0.01% SDS, or 0.05% Congo red at 25°C for 10 days. Relative growth inhibition caused by chemical

stress was calculated using the following equation: $[(Sc - St)/(Sc - d)] \times 100$, where Sc and St denote the areas of the unstressed (control) and stressed colonies, respectively, and *d* is the area of mycelia plug used to initiate cultures (36).

**CR formation and inflation assays.** CR formation and ring cell inflation were measured at multiple time points at 25℃ for 3 days after inoculating approximately $10^4$ conidia of the wild type and its mutants on WA plates. After adding approximately 1,500 *C. elegans* to each plate, the number of CRs was counted at 12, 24, and 36 h. Water at 55℃ was added to the plates to stimulate CR inflation (25), and the numbers of uninflated and inflated CRs were counted to determine the CR inflation rate.

**Microscopy, scanning electron microscopy, and TEM analyses.** After culturing the wild-type and mutant strains on WA plates for 5 days at 25℃, about 1,500 *C. elegans* worms were added to induce CR formation. Agar pieces were sliced and placed on a slide to observe CR morphology under a $100\times$ oil immersion lens. For confocal microscope observation, conidia were cultured in 1% horse serum for 5 days at 25℃ to produce mycelia with CRs (25). The vacuoles of CRs and mycelia were stained with neutral red (Solarbio, Beijing, China) or CellTracker blue CMAC (Invitrogen, OR, USA) for 15 min. The stained specimens were washed three times with phosphate-buffered saline (PBS; pH 7.0) before observation with a confocal microscope (Leica TCSSP5, Germany).

For TEM examination, the fungal strains were cultured on PDA medium covered with cellophane membrane for 5 days at 25℃. Mycelial mass was collected, fixed overnight at 4℃ in 50 mM sodium phosphate buffer (pH 7.2) containing 2.5% glutaraldehyde, and washed three times with PBS (pH 7.2). The samples were dehydrated using increasing concentrations of ethanol (40 to 100%) and embedded in resin for sectioning. Ultrathin samples were treated with 2% uranium acetate followed by lead citrate before imaging via a transmission electron microscope (Hitachi, Japan).

**Statistical analysis.** All the experiments were performed with at least three biological replicates and repeated ≥5 times for each group, and the resulting data are presented as means ± standard deviations. The unpaired Student's *t* test was used to compare the differences between groups, and *P* values of <0.05 were considered statistically significant. All statistical analyses were conducted using GraphPad Prism 8.

**Data availability.** The whole-genome shotgun sequence of wild-type *D. dactyloides* strain CGMCC3.20198 was deposited with GenBank and assigned accession number JAGTWJ000000000 (BioSample SAMN18837316; BioProject number PRJNA723920).

## SUPPLEMENTAL MATERIAL

Supplemental material is available online only.
**SUPPLEMENTAL FILE 1**, MP4 file, 8.4 MB.
**SUPPLEMENTAL FILE 2**, MP4 file, 6.8 MB.
**SUPPLEMENTAL FILE 3**, MP4 file, 12 MB.
**SUPPLEMENTAL FILE 4**, MP4 file, 9.7 MB.
**SUPPLEMENTAL FILE 5**, MP4 file, 7.6 MB.
**SUPPLEMENTAL FILE 6**, PDF file, 1.4 MB.

## ACKNOWLEDGMENTS

We thank Di An (College of Life Science, Nankai University) for helping obtain and interpret TEM images. This study was supported by grants from the National Natural Science Foundation Program of China (grants 31770065 and 32020103001).

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
