## [Reviewer comments · Microbiology Spectrum]

Microbiology Spectrum

SNARE protein DdVam7 of nematode-trapping fungus *Drechlerella dactyloides* regulates vegetative growth, conidiation and predatory process via vacuole assembly

Yue Chen, Jia Liu, Yani Fan, Meichun Xiang, Seogchan Kang, Dongsheng Wei, and Xingzhong Liu

Corresponding Author(s): Xingzhong Liu, Nankai University

Review Timeline:

Submission Date:	May 20, 2022
Editorial Decision:	June 20, 2022
Revision Received:	September 14, 2022
Accepted:	September 30, 2022

Editor: Chengshu Wang

Reviewer(s): The reviewers have opted to remain anonymous.

Transaction Report:

DOI: <https://doi.org/10.1128/spectrum.01872-22>

June 20, 2022

Prof. Xingzhong Liu
Nankai University
College of Life Science
No.94 Weijin Road, Nankai District
Tianjin 300071
China

Re: Spectrum01872-22 (SNARE protein DdVam7 of nematode-trapping fungus *Drechlerella dactyloides* regulates vegetative growth, conidiation and predatory process via vacuole assembly)

Dear Prof. Xingzhong Liu:

Link Not Available

Sincerely,

Chengshu Wang

Journals Department
Reviewer comments:

Reviewer #1 (Comments for the Author):

The manuscript by Yue Chen et al. describes the role of the SNARE protein DdVam7 in the constricting ring-forming nematode-trapping fungus of *D. dactyloides*. The authors describe that deletion of the gene encoding DdVam7 leads to suppressed vegetative growth and conidiation, impaired trap formation and ring cell inflation, as well as decreased nematode-trapping ability. They further characterized the vacuole of the mutant strain as small and fragmented in hypha and uninflated ring cells and observed the inability of the mutant to form the large vacuole in inflated ring cells. DdVam7 fused with RFP localized in vacuoles. The study concludes a role of DdVam7 in vegetative growth, conidiation and the predatory process.

Overall, this is an interesting and well-composed study describing the role of a SNARE protein in vacuole assembly and constricting ring formation. It adds to the list of proteins described to play a role in trap formation in NTF. The movies in the supplemental information provide a nice demonstration of the described processes. However, the study depicts a rather basic description of the phenotype of one mutant, lacking a deeper mechanistic analysis.

1) Main concern: localization of DdVam7 was shown with a C-terminal GFP fusion that was described to accumulate in vacuoles. Co-localization with CMAC was only shown in spores (Figure 5). This localization is different from the ones described for other Vam7 orthologous in the literature e.g. in *F. oxysporum* (<https://doi.org/10.1111/1462-2920.14585>) or *Colletotrichum fruticola* (<https://doi.org/10.3389/fmicb.2021.736066>). Here, Vam7 is reported to localize as ring-shaped structures co-localizing with the vacuole membrane indicating localization at the vacuole membrane. How do the authors explain this discrepancy? As previous studies performed N-terminal GFP fusions of the Vam7 protein and this study performed a C-terminal fusion I suggest confirming this observation with an N-terminal GFP fusion. Additionally, the authors should analyze the localization in the deletion background to show that the tagged version is functional and able to rescue the mutant phenotypes e.g., in growth assays. If the protein still localizes in the vacuoles, the localization should be shown under different growth conditions.

2) As part of a deeper characterization of the role of Vam7 in the virulence of the fungus, I suggest looking into the importance of the PX motif and SNARE domain the protein contains. These are described to be essential for normal localization and biological functions of Vam7. The SNARE domain was described as essential in *C. fruticola* while the PX motif was indicated to play a minor role in virulence in this fungus (<https://doi.org/10.3389/fmicb.2021.736066>). The authors should generate mutants lacking just the respective domain and compare their phenotype to the full deletion to see if one domain might be essential for ring closure.

3) Hypotheses about the mechanistic background of the formation of the large vacuole in CRs and the role of SNARE proteins would improve the discussion. So far, the discussion mainly summarizes some of the relevant literature and doesn't really stress how SNAREs might play a distinct role in constricting-ring forming NTF distinguishing the study from the other characterizations of Vam7 in fungi.

Lines 138-139: How do the authors explain the incomplete complementation of the CR formation?

Lines 140-141 state the number of traps for for DdVam7C and wild type as ca. 1201 vs ca. 1296 (36 hours after introducing *C. elegans*) but the graph in Fig. 3 B does not represent these numbers as there seems to be a much bigger difference. It is not clear from the text if the approximate number given is the median of the 3 different complementation strains?

Lines 157-159: additional time-lapse images or videos would make the defect in vacuolar fusion clearer.

Figure 1 and 2 can be combined. Figure 1 could be reduced by moving 1 C & E to the supplements.

Figures 1 B: how do the authors explain the increase of expression at 24 h after it went down at 16 hours?

Figure 5 and 6 could be combined.

Reviewer #2 (Comments for the Author):

The manuscript by Yew Chen and co-workers demonstrate that the SNARE protein DdVam7 plays an important role in *D. dactyloides*, including vegetative growth, conidiation, and trap formation and function. The authors generated Δ DdVam7 mutants that reduced mycelial growth, conidiation, and trap formation. In addition, the mutants are defected in CR inflation which loses the ability to capture *C. elegans*. These defective phenotypes can be rescued in the complementary strains. The authors show that the DdVam7 proteins are expressed in vacuoles, and the Δ DdVam7 mutants contain abnormal vacuoles in both vegetative hyphae and traps. I list some questions and suggestions below.

1. Can author elaborate more about why choosing Vam7? How many SNARE proteins in *D. dactyloides*?
2. It looks like all the trap-related phenotype can be attributed to the poor growth of the mutant, but not specifically related to morphogenesis. Vam7 still formed constricting ring and the ring can also contract after longer time. It is confusing that the author wrote "DdVam7 is required for CR morphogenesis" but CR was observed in Fig 3. And Fig. 4.
3. Can author test the phenotype of the Vam7 mutant and compare with wild-type that has comparable growth? For example, 3 weeks old culture of vam7 showed the same growth as 5 day wild-type, then I would like to see the phenotype of CR development and inflation of a 3 week-old mutant and compared that with the 5 day-old wild-type.
4. The localization of DdVam7 has been demonstrated by fusion proteins. However, why the fusion proteins express in the lumen of vacuoles instead of the membrane of vacuoles? The authors claim that DdVam7 may involve in vacuoles fusion. If that's the case, localization of Vam7 should be on the vacuolar membrane.
5. There is a latency between ring cell inflation and large vacuole developed, which raise a question that what is the function of vacuole in CR. In addition, the DdVam7 mutants contain abnormal vacuole and still show inflated CR in figure 4E and 6D, which

is contradictory that vacuole fusion is involved in CR inflation. The authors need to provide more evidence that the vacuole fusion plays a critical role in CR inflation to support their finding that the defective trap function of DdVam7 mutants is due to failing vacuole fusion.

Staff Comments:

Preparing Revision Guidelines

Please return the manuscript within 60 days; if you cannot complete the modification within this time period, please contact me. If you do not wish to modify the manuscript and prefer to submit it to another journal, please notify me of your decision immediately so that the manuscript may be formally withdrawn from consideration by Microbiology Spectrum.

The manuscript by Yue Chen et al. describes the role of the SNARE protein DdVam7 in the constricting ring-forming nematode-trapping fungus of *D. dactyloides*. The authors describe that deletion of the gene encoding DdVam7 leads to suppressed vegetative growth and conidiation, impaired trap formation and ring cell inflation, as well as decreased nematode-trapping ability. They further characterized the vacuole of the mutant strain as small and fragmented in hypha and uninflated ring cells and observed the inability of the mutant to form the large vacuole in inflated ring cells. DdVam7 fused with RFP localized in vacuoles. The study concludes a role of DdVam7 in vegetative growth, conidiation and the predatory process. Overall, this is an interesting and well-composed study describing the role of a SNARE protein in vacuole assembly and constricting ring formation. It adds to the list of proteins described to play a role in trap formation in NTF. The movies in the supplemental information provide a nice demonstration of the described processes. However, the study depicts a rather basic description of the phenotype of one mutant, lacking a deeper mechanistic analysis.

- 1) Main concern: localization of DdVam7 was shown with a C-terminal GFP fusion that was described to accumulate in vacuoles and co-localization with CMAC was performed (Figure 5). This localization is different from the ones described for other Vam7 orthologous in the literature e.g. in *F. oxysporum* (<https://doi.org/10.1111/1462-2920.14585>) or *Colletotrichum fruticola* (<https://doi.org/10.3389/fmicb.2021.736066>). Here, Vam7 is reported to localize as ring-shaped structures co-localizing with the vacuole membrane indicating localization at the vacuole membrane. How do the authors explain this discrepancy?
As previous studies performed N-terminal GFP fusions of the Vam7 protein and this study performed a C-terminal fusion I suggest confirming this observation with an N-terminal GFP fusion. Additionally, the authors should analyze the localization in the deletion background to show that the tagged version is functional and able to rescue the mutant phenotypes e.g., in growth assays. If the protein still localizes in the vacuoles, the localization should be shown under different growth conditions.
- 2) As part of a deeper characterization of the role of Vam7 in the virulence of the fungus, I suggest looking into the importance of the PX motif and SNARE domain the protein contains. These are described to be essential for normal localization and biological functions of Vam7. The SNARE domain was described as essential in *C. fruticola* while the PX motif was indicated to play a minor role in virulence in this fungus (<https://doi.org/10.3389/fmicb.2021.736066>). The authors should generate mutants lacking just the respective domain and compare their phenotype to the full deletion to see if one domain might be essential for ring closure.
- 3) Hypotheses about the mechanistic background of the formation of the large vacuole in CRs and the role of SNARE proteins would improve the discussion. So far, the discussion mainly summarizes some of the relevant literature and doesn't really stress how SNAREs might play a distinct role in constricting-ring forming NTF distinguishing the study from the other characterizations of Vam7 in fungi.

Lines 138-139: How do the authors explain the incomplete complementation of the CR formation?

Lines 140-141 state the number of traps for for *DdVam7C* and wild type as ca. 1201 vs ca. 1296 (36 hours after introducing *C. elegans*) but the graph in Fig. 3 B does not represent these numbers as there seems to be a much bigger difference. It is not clear from the text if the approximate number given is the median of the 3 different complementation strains?

Lines 157-159: additional time-lapse images or videos would make the defect in vacuolar fusion clearer.

Figure 1 and 2 can be combined. Figure 1 could be reduced by moving 1 C & E to the supplements.

Figures 1 B: how do the authors explain the increase of expression at 24 h after it went down at 16 hours?

Figure 5 and 6 can be combined.

Point-to-point response to reviewers' comments.

We thank the reviewers for their encouraging comments and helpful suggestions. Our responses are in blue and all changes in the revised manuscript text file with track changes.

Reviewer comments:

Reviewer #1 (Comments for the Author):

The manuscript by Yue Chen et al. describes the role of the SNARE protein DdVam7 in the constricting ring-forming nematode-trapping fungus of *D. dactyloides*. The authors describe that deletion of the gene encoding DdVam7 leads to suppressed vegetative growth and conidiation, impaired trap formation and ring cell inflation, as well as decreased nematode-trapping ability. They further characterized the vacuole of the mutant strain as small and fragmented in hypha and uninflated ring cells and observed the inability of the mutant to form the large vacuole in inflated ring cells. DdVam7 fused with RFP localized in vacuoles. The study concludes a role of DdVam7 in vegetative growth, conidiation and the predatory process.

Overall, this is an interesting and well-composed study describing the role of a SNARE protein in vacuole assembly and constricting ring formation. It adds to the list of proteins described to play a role in trap formation in NTF. The movies in the supplemental information provide a nice demonstration of the described processes. However, the study depicts a rather basic description of the phenotype of one mutant, lacking a deeper mechanistic analysis.

R: Thanks for the reviewer's positive comments. There are a number of SNARE proteins in fungi. This is the first investigation on SNARE proteins in nematode-trapping fungi.

1) Main concern: localization of DdVam7 was shown with a C-terminal GFP fusion that was described to accumulate in vacuoles. Co-localization with CMAC was only shown in spores (Figure 5). This localization is different from the ones described for other Vam7 orthologous in the literature e.g., in *F. oxysporum* (<https://doi.org/10.1111/1462-2920.14585>) or *Colletotrichum fructicola* (<https://doi.org/10.3389/fmicb.2021.736066>). Here, Vam7 is reported to localize as ring-shaped structures co-localizing with the vacuole membrane indicating localization at the vacuole membrane. How do the authors explain this discrepancy?

As previous studies performed N-terminal GFP fusions of the Vam7 protein and this study performed a C-terminal fusion. I suggest confirming this observation with an N-terminal GFP fusion. Additionally, the authors should analyze the localization in the deletion background to show that the tagged version is functional and able to rescue the mutant phenotypes e.g., in growth assays. If the protein still localizes in the vacuoles, the localization should be shown under different growth conditions.

R: This is a very valuable suggestion. To confirm the intracellular localization of DdVam7, we constructed a strain ($\Delta DdVam7/RFP-DdVam7$) with a RFP-tag at the N-

terminus of DdVam7 in the $\Delta DdVam7$ mutant genetic background. We characterized phenotypes of the $\Delta DdVam7/RFP-DdVam7$ and showed the localization of DdVam7 in the apical and basal region of hyphae growing on PDA or WA plates covered with cellophane membrane. The new acquired data are shown in Fig.4A, Fig. S6, and Response Fig. 1 (below) and the manuscript has been revised accordingly.

In $\Delta DdVam7/RFR-DdVam7$, vegetative growth and CR formation were largely restored (Fig. S6 B-E), and its ring cells' inflation was completely restored (Fig. S6F and G). Hyphae of $\Delta DdVam7/RFP-DdVam7$ were incubated on PDA or WA plates covered with cellophane membrane and collected for observation. DdVam7 fused with RFP accumulated in punctate structures of apical region of hyphae and in large organelles of basal region of hyphae. Colocalization of RFP signal with that of 7-amino-4-chloromethylcoumarin (CMAC), a fluorescent dye that stains the interior of vacuoles, indicated that DdVam7 was located in vacuoles (Fig. 4A and Response Fig. 1).

Similar results have been reported in *Fusarium graminearum*, FgVam7 is observed mainly in the vacuoles of the vegetative hyphae (1). However, these results do not exclude the possibility that FgVam7 and DdVam7 are also located on the vacuolar membrane. It is possible that the SNARE protein Vam7 may have a different regulatory mechanism in different fungi, playing distinct roles in different stages of exocytosis or endocytosis, such as membrane fusion or vacuole assembly and sorting.

Fig. S6 Phenotypic characteristics of the $\Delta DdVam7/RFP-DdVam7$. (A) Construction of $\Delta DdVam7/RFP-DdVam7$ strains. M1: DNA marker D2000; M2: DNA marker 1 Kb; H: partial hygromycin resistance gene amplified using primers HYG-F and HYG-R; G: partial geneticin resistance gene amplified using primers NEO-F and NEO-R; RV: PCR results amplified using primers SR-F and SR-R. The regions amplified via PCR and the locations of individual primers were marked in upper panel. (B) Colony diameters of wild type and $\Delta DdVam7/RFP-DdVam7$ cultured on PDA plates for 10 days. ** $P < 0.01$ ($n = 5$) (C) Colony morphology of WT and $\Delta DdVam7/RFP-DdVam7$ cultured on PDA plates for 10 days. (D) The numbers of CRs formed at 36 hours after inoculating *C. elegans*. ns = not significant ($n = 5$). (E) Light micrographs showing CR formation capacity at 36 hours after inoculating *C. elegans*. The lower panel is close-up view

indicated by dotted black box. Shown are representatives of at least five images. Scale bars = 50 μm . (F) The percentages of inflated CRs at 72 hours after inoculating *C. elegans*. ns = not significant (n = 6). (G) Light micrographs showing ring cell inflation. The lower panel is close-up view indicated by dotted black box. Scale bars = 50 μm .

Fig. 4A Micrographs showing that DdVam7 localizes in vacuoles in hyphae. Scale bar = 5 μm . Hyphae of $\Delta DdVam7/RFP-DdVam7$ were incubated on PDA plates covered with cellophane membrane.

Response Fig. 1 Micrographs showing that DdVam7 localizes in vacuoles in hyphae. Scale bar = 5 μm . Hyphae of $\Delta DdVam7/RFP-DdVam7$ were incubated on WA plates covered with cellophane membrane.

2) As part of a deeper characterization of the role of Vam7 in the virulence of the fungus, I suggest looking into the importance of the PX motif and SNARE domain the protein contains. These are described to be essential for normal localization and biological functions of Vam7. The SNARE domain was described as essential in *C. fruticola* while the PX motif was indicated to play a minor role in virulence in this fungus

(<https://doi.org/10.3389/fmicb.2021.736066>). The authors should generate mutants lacking just the respective domain and compare their phenotype to the full deletion to see if one domain might be essential for ring closure.

R: We agree with the reviewer's comments. To explore the functions of the PX and SNARE domains, *DdVam7*^{ΔPX} and *DdVam7*^{ΔSNARE} deletion domain constructs were generated and transformed into the $\Delta DdVam7$. We compared the phenotype of domain deletion strains to the wild type, $\Delta DdVam7$ and *DdVam7*^C. The new acquired data are shown in Fig. 5 and Fig. S7 (below) and the manuscript has been revised accordingly.

We generated *DdVam7*^{ΔPX} and *DdVam7*^{ΔSNARE} deletion domain constructs and transformed each construct into the $\Delta DdVam7$ (Fig. 5A; Fig. S7). The colonies of the domain deletion strains showed moderate growth rates between wild type and $\Delta DdVam7$. After 10 days of growth on PDA, the average colony diameter of wild type and $\Delta DdVam7$ were ca. 35 mm and ca. 13 mm, while that of $\Delta DdVam7/DdVam7$ ^{ΔPX} and $\Delta DdVam7/DdVam7$ ^{ΔSNARE} were ca. 22 mm and ca. 18 mm (Fig. 5B and C). The PX and SNARE domain-truncated strains displayed impaired CR formation capacity similar to $\Delta DdVam7$ at 24 hours after inoculating *C. elegans* (Fig. 5D and E). After 36 hours, the domain deletion strains could produce more traps than $\Delta DdVam7$. The trap numbers per cm² of $\Delta DdVam7/DdVam7$ ^{ΔPX}, $\Delta DdVam7/DdVam7$ ^{ΔSNARE} and $\Delta DdVam7$ were ca. 55, ca. 44 and ca. 24 (Fig. 5D and E). Comparison with wild type and *DdVam7*^C, the CRs of the SNARE domain deletion strain failed to inflate to capture nematodes at 36 hours after inoculating *C. elegans* and also failed to inflate even by heat stimulation at 72 hours after inoculating *C. elegans*, which was similar with the $\Delta DdVam7$ (Fig. 5 E-G). On the contrary, the PX domain deletion strain could inflate to capture nematodes like wild type and *DdVam7*^C at 36 hours after inoculating *C. elegans* (Fig. 5E). At 72 hours after inoculating *C. elegans*, 97% CRs of the PX domain deletion strain could inflate by heat stimulation (Fig. 5F and G). These findings demonstrated that both SNARE and PX domains of *DdVam7* participated in the regulation of growth and trap formation, and the SNARE domain play an important role in ring cell inflation.

Fig. S7 Domain deletion. (A) Schematic diagram of the domain deletion strategy. (B, C) Verification of domain disruption using PCR. M1: DNA marker D2000; M2: DNA marker 1 Kb; H: partial hygromycin resistance gene amplified using primers HYG-F and HYG-R; G: partial geneticin resistance gene amplified using primers NEO-F and NEO-R; S: PCR results amplified using primers SNA-F and SNA-R; P: PCR results amplified using primers PX-F and PX-R. The regions amplified via PCR and the locations of individual primers were marked in (A).

FIG 5 The PX and SNARE domain is important for the function of DdVam7. (A) Schematic showing domain deletions of DdVam7. (B) Colony diameters of WT, Δ DdVam7, DdVam7^C and domain deletion strains. (C) Colony morphology of WT, Δ DdVam7, DdVam7^C and domain deletion strains cultured on PDA plates for 10 days. (D) Bar chart showing the numbers of CRs formed at 24 and 36 hours after inoculating *C. elegans*. ***P < 0.001, ns = not significant (n = 5). (E) Light micrographs showing CR formation capacity at 36 hours after inoculating *C. elegans*. The lower panel is close-up view indicated by dotted black box in upper panel. Shown are representatives of at least five images. Scale bars = 50 μ m. (F) Bar chart showing the percentages of inflated CRs at 72 hours after inoculating *C. elegans*. ***P < 0.001, ns = not significant (n = 6). (G) Light micrographs showing ring cell inflation at 72 hours after inoculating *C. elegans*. The lower panel shows close-up views of the areas indicated by dotted black box in the upper panel. Shown are representatives of at least five images. Scale bars = 50 μ m.

3) Hypotheses about the mechanistic background of the formation of the large vacuole in CRs and the role of SNARE proteins would improve the discussion. So far, the discussion mainly summarizes some of the relevant literature and doesn't really stress how SNAREs might play a distinct role in constricting-ring forming NTF distinguishing the study from the other characterizations of Vam7 in fungi.

R: Indeed, we have a relatively broad understanding of the morphological properties of CRs. There is little known about the molecular mechanisms underlying the rapid inflation of ring cells and, in particular, the mechanistic background of the formation of large vacuoles. We have revised this issue in the Discussion as following:

“Much of our understanding of the role of vacuole assembly in rapid and large cell volume change has been derived from research on guard cells. In *Vicia faba*, guard cells can also undergo large changes in cell volume in a short period of time during stomatal movement. It has been implicated that the small vacuoles can fuse with each other or with the bigger vacuoles to generate large vacuoles, and vacuole volume change has a major contribution to the change of guard cell volume (2).” (Lines 263-268 in revised manuscript)

“MoVam7 in *M. oryzae* is involved in maintaining the shapes of vacuoles and contributed to the regulation of cell wall integrity, endocytosis, conidiation and pathogenicity (3). FgVam7 participated in vacuolar maintenance and endocytosis, consequently regulating vegetative growth, conidiation and conidial germination, sexual reproduction and virulence in *F. graminearum* (1). FolVam7 of *F. oxysporum* is involved in intracellular trafficking, thereby regulating vegetative growth, asexual development, conidial morphology and plant infection (4). In *Camellia oleifera*, CfVam7 participated in vacuole fusion, appressorium formation, and pathogenicity (5).” (Lines 228-236 in revised manuscript)

4) Lines 138-139: How do the authors explain the incomplete complementation of the CR formation?

R: We thank the reviewer for pointing out this issue. In the initial experiment, we obtained 17 complementary strains, compared their growth, conidiation and ring formation, and selected three strains with better complementation. We thought that the sites of T-DNA integration (i.e., positional effect) might affect the gene expression, resulting in differential complementation among transformants. As we can see in Fig. 1D in the revised manuscript, *DdVam7* expression of three complementary strains were not totally complementary to wild type, which were 85%, 91% and 94% of the wild strain, respectively. Furthermore, high expression of *DdVam7* was required during CR formation (within 8 to 12 hours after inoculating *C. elegans*, Fig. 1B in the revised manuscript). Randomly inserted and expressed *DdVam7* may not sufficiently reach the desired expression level for CR formation. As we can see in Fig. 2A and B in the revised manuscript, the number of CRs in $\Delta DdVam7$ was only 2.1% and 1.5% of that in the wild type at 24 and 36 hours after inoculating *C. elegans*, respectively. However, the CRs of complementary strains was 72.7% (24 hours after inoculating *C. elegans*) and 74.3% (36 hours after inoculating *C. elegans*) of that in wild type. Obviously, the

complementation of *DdVam7* had almost restored the ability of CR formation.

5) Lines 140-141 state the number of traps for *DdVam7^C* and wild type as ca. 1201 vs ca. 1296 (36 hours after inoculating *C. elegans*) but the graph in Fig. 3 B does not represent these numbers as there seems to be a much bigger difference. It is not clear from the text if the approximate number given is the median of the 3 different complementation strains?

R: Yes, we recalculated the median of traps produced by 3 different complementary strains. The number of traps for *DdVam7^C* and wild type as ca. 963 vs ca. 1296 (36 hours after inoculating *C. elegans*). We have corrected this mistake in the revised manuscript in line 142.

6) Lines 157-159: additional time-lapse images or videos would make the defect in vacuolar fusion clearer.

R: Good suggestion but difficult to be done. We were unable to show the defect in vesicle fusion process through video due to the facilities limitation to capture this process. As we can see in Fig. 4C, the transmission electron micrographs clearly show that small vacuoles failed to fusion completely in $\Delta DdVam7$.

7) Figure 1 and 2 can be combined. Figure 1 could be reduced by moving 1 C & E to the supplements.

R: As suggested, we have combined Fig. 1 and 2 as Fig. 1 in the revised manuscript and moved Fig. 1C and E into the Supplementary Information as Fig. S3.

8) Figures 1 B: how do the authors explain the increase of expression at 24 h after it went down at 16 hours?

R: It is a good point raised. As we can see in Fig. S2, a lot of CRs form at 8-12 hours after inoculation of *C. elegans*. These CRs require several hours (ca. 8 hours) of maturation to be capable of inflation and most of CRs are able to inflate at 24 hours and capture nematodes. *DdVam7* expression is high at 8-12 hours, decreases at 16-20 hours, and then increases at 24 hours, suggesting that *DdVam7* is involved in the CR formation and inflation. Deletion of *DdVam7* impairs CR formation and ring cell inflation, which further supports the role of *DdVam7*. We have explained this issue in lines 109-113.

9) Figure 5 and 6 could be combined.

R: Yes, we have combined Figure 5 and 6 as Figure 4 in the revised manuscript.

Reviewer #2 (Comments for the Author):

The manuscript by Yew Chen and co-workers demonstrate that the SNARE protein DdVam7 plays an important role in *D. dactyloides*, including vegetative growth, conidiation, and trap formation and function. The authors generated $\Delta DdVam7$ mutants that reduced mycelial growth, conidiation, and trap formation. In addition, the mutants are defected in CR inflation which loses the ability to capture *C. elegans*. These defective phenotypes can be rescued in the complementary strains. The authors show that the DdVam7 proteins are expressed in vacuoles, and the $\Delta DdVam7$ mutants contain abnormal vacuoles in both vegetative hyphae and traps. I list some questions and suggestions below.

R: Thanks for the positive comments.

1) Can author elaborate more about why choosing Vam7? How many SNARE proteins in *D. dactyloides*?

R: In *D. dactyloides*, approximately 16 putative SNARE proteins (containing SNARE domain) could be predicted in the genome (Table S1). SNARE protein Vam7 has been studied in yeast and some filamentous fungi and has been shown to be involved in vacuole fusion. Our previous study showed that deletion of *DdaSte12*, a transcription factor working downstream of the MAPK signaling pathway in *D. dactyloides* caused malformed vacuoles in the ring cells and disabled ring cell inflation. Deletion of *DdaSTE12* also altered the expression levels of genes related to SNARE domain attachment proteins involved in vesicular membrane fusion events (6). Especially, there is a large centrally-located vacuole formed during CR ring cell inflation. Therefore, we hypothesize that DdVam7 should be involved in the vacuole assembling in CR formation in *D. dactyloides*.

Table S1. List of putative SNARE proteins in *D. dactyloides*.

Locus name	Annotated function	Length (a.a)	SNARE domain position
1295_g_dac	Syntaxin/t-SNARE Sso1	309	208-271
2995_g_dac	Syntaxin/t-SNARE Sso1	470	371-433
3420_g_dac	Synaptobrevin/Vesicle-trafficking protein Sec22	215	133-212
3592_g_dac	Synaptobrevin/VAMP-like protein/v-SNARE SNC1	237	141-227
3734_g_dac	Qc-SNARE Vam7	361	297-340
4406_g_dac	Golgi SNAP receptor complex member 2 (GOSR2)	329	241-302

5041_g_dac	t-SNARE Syntaxin5 (Syn5)	321	266-317
5412_g_dac	v-SNARE VTI1 family protein	226	21-98
5465_g_dac	t-SNARE Syntaxin-2	272	199-239
621_g_dac	t-SNARE Syntaxin-16	419	236-296
65_g_dac	t-SNARE Syntaxin-8	306	219-286
7439_g_dac	SNARE Syntaxin-18	352	265-323
7538_g_dac	Synaptobrevin/VAMP-like protein/v-SNARE SNC1	119	29-88
8115_g_dac	t-SNARE Syntaxin-6	240	147-208
8656_g_dac	Golgi SNAP receptor complex member 1 (GOSR1)	225	141-202
8744_g_dac	Synaptobrevin/R-SNARE YKT6	207	145-205

2) It looks like all the trap-related phenotype can be attributed to the poor growth of the mutant, but not specifically related to morphogenesis. Vam7 still formed constricting ring and the ring can also contract after longer time. It is confusing that the author wrote "DdVam7 is required for CR morphogenesis" but CR was observed in Fig 3. And Fig. 4.

R: Thanks for this crucial comment. Deletion of *DdVam7* severely impaired CR formation and ring cell inflation, and the diameter of $\Delta DdVam7$ CRs was significantly smaller than that of wild type, which implies that DdVam7 is probably required for CR morphogenesis. To avoid inexactly statement, we revised the description as "Deletion of DdVam7 impaired CR formation and ring cell inflation" in the revised manuscript in line 137.

3) Can author test the phenotype of the Vam7 mutant and compare with wild-type that has comparable growth? For example, 3 weeks old culture of *vam7* showed the same growth as 5 day wild-type, then I would like to see the phenotype of CR development and inflation of a 3-week-old mutant and compared that with the 5 day-old wild-type.

R: Good suggestions. To test the phenotype of the $\Delta DdVam7$ and compare with wild type that has comparable growth, mycelia plug of these two strains (6 mm in diameter) were inoculated on WA medium at 25°C. The newly obtained data are shown below and in Fig. S5 of the revised manuscript.

We found that 25-day-old culture of $\Delta DdVam7$ had a colony diameter of approximately 22 mm, comparable to that of the 3-day-old wild-type (Fig. S5A and B). The trap formation in $\Delta DdVam7$ is significantly reduced. The number of traps in

$\Delta DdVam7$ was only 8.9% (24 hours after challenge of *C. elegans*) and 9.5% (36 hours after inoculating *C. elegans*) of that in wild type (Fig. S5C and D). CRs of $\Delta DdVam7$ failed to inflate by heat stimulation at 72 hours after inoculating *C. elegans*, whereas 98% CRs of wild type inflated. However, a few CRs of $\Delta DdVam7$ could inflate by nematodes or hot water stimulation after one week of induction (Fig. S5E and F).

Fig. S5 CR formation and inflation of the $\Delta DdVam7$ and wild type that has comparable growth. (A) Colony diameters of wild type (WT, 3-days-old culture) and $\Delta DdVam7$ (25-day-old culture). ns = not significant (n = 5). (B) Colony morphology of WT and $\Delta DdVam7$ cultured on WA plates for 3 days and 25 days respectively. (C) The numbers of CRs formed at 24 and 36 hours after inoculating *C. elegans*. ***P < 0.001 (n = 6). (D) Light micrographs showing CR formation capacity. Blue arrows indicate the CRs. The right panel is close-up view indicated by dotted black box. Shown are representatives of at least five images. Scale bars = 50 μ m. (E) The percentages of inflated CRs at 72 hours and 1 week after inoculating *C. elegans* into cultures of the wild type (WT) and $\Delta DdVam7$. ***P < 0.001, NF = not found (n = 6). (F) Light micrographs showing ring cell inflation. Red arrows indicate inflated CRs. Scale bars = 50 μ m.

4) The localization of DdVam7 has been demonstrated by fusion proteins. However, why the fusion proteins express in the lumen of vacuoles instead of the membrane of vacuoles? The authors claim that DdVam7 may involve in vacuoles fusion. If that's the case, localization of Vam7 should be on the vacuolar membrane.

R: In *F. oxysporum* or *C. fructicola*, Vam7 is reported to localize as ring-shaped structures co-localizing with the vacuole membrane indicating localization at the vacuole membrane (4, 5). Here, we showed that DdVam7 accumulated in punctate structures of apical region of hyphae and in large organelles of basal region of hyphae (Fig. 4A). Similar results have been reported in *F. graminearum*, FgVam7 is observed mainly in the vacuoles of the vegetative hyphae (1). However, these results do not exclude the possibility that FgVam7 and DdVam7 are also located on the vacuolar membrane and may involve in vacuole fusion. We deduce that the SNARE protein Vam7 may have a different regulatory mechanism in different fungi, playing distinct roles in different stages of exocytosis or endocytosis, such as membrane fusion or vacuole assembly and sorting. We have discussed this point in lines 238-248.

5) There is a latency between ring cell inflation and large vacuole developed, which raise a question that what is the function of vacuole in CR. In addition, the *DdVam7* mutants contain abnormal vacuole and still show inflated CR in figure 4E and 6D, which is contradictory that vacuole fusion is involve in CR inflation. The authors need to provide more evident that the vacuole fusion plays a critical role in CR inflation to support their founding that the defective trap function of DdVam7 mutants is due to failing vacuole fusion.

R: This is a tough question to be answered. Generally, the cell shape changes are mainly driven by vacuole. During the CR ring inflation, a large central vacuole formed to adopt the volume increase in the ring cell. This may be because vacuole biogenesis is energetically less costly than generating new cytoplasmic content (7). Therefore, we believe that the main function of vacuoles in ring cells is to enable ring cells to achieve rapid morphological changes in a short period of time.

CR inflation is completed in a very short time, which is resulted in technical difficult to determine whether the vacuole fusion is involved in CR inflation. What can be observed is that vacuole reassembles during inflation, forming large and round vacuoles that occupy almost the entire ring cells (Fig. 4D), perhaps generating enough mechanical force to firmly trap the nematode. In $\Delta DdVam7$, the vacuoles of uninflated CRs are malformed, and no large and round vacuoles were formed in inflated ring cells. Vacuoles are crucial for nutrient transport in fungi, and abnormal vacuoles may interfere with the CR maturation, resulting in only a few CRs that can inflate at 1 week after inoculating *C. elegans*. However, lack large vacuoles in these inflated CRs suggests the potential role of Vam7 in the formation of large vacuoles. The mechanisms underlying the formation of large vacuoles and the molecules directly involved remain to be further explored. We have revised the text in the Discussion section accordingly in lines 269-279.

Reference

1. Zhang HF, Li B, Fang Q, Li Y, Zheng XB, Zhang ZG. 2016. SNARE protein FgVam7 controls growth, asexual and sexual development, and plant infection in *Fusarium graminearum*. *Mol Plant Pathol* 17:108-119.
2. Gao XQ, Li CG, Wei PC, Zhang XY, Chen J, Wang XC. 2005. The dynamic changes of tonoplasts in guard cells are important for stomatal movement in *Vicia faba*. *Plant Physiol* 139:1207-1216.
3. Dou XY, Wang Q, Qi ZQ, Song WW, Wang W, Guo M, Zhang HF, Zhang ZG, Wang P, Zheng XB. 2011. MoVam7, a conserved SNARE involved in vacuole assembly, is required for growth, endocytosis, ROS accumulation, and pathogenesis of *Magnaporthe oryzae*. *PLoS One* 6:e16439.
4. Li B, Gao Y, Mao HY, Borkovich KA, Ouyang SQ. 2019. The SNARE protein FolVam7 mediates intracellular trafficking to regulate conidiogenesis and pathogenicity in *Fusarium oxysporum* f. sp. *lycopersici*. *Environ Microbiol* 21:1462-2920.
5. Li S, Zhang S, Li B, Li H. 2021. The SNARE protein CfVam7 is required for growth, endoplasmic reticulum stress response, and pathogenicity of *Colletotrichum fructicola*. *Front Microbiol* 12:736066.
6. Fan YN, Zhang WW, Chen Y, Xiang MC, Liu XZ. 2021. DdaSTE12 is involved in trap formation, ring inflation, conidiation, and vegetative growth in the nematode-trapping fungus *Drechlerella dactyloides*. *Appl Microbiol Biotechnol* 105:7379-7393.
7. Veses V, Richards A, Gow NA. 2008. Vacuoles and fungal biology. *Curr Opin Microbiol* 11:503-510.

September 30, 2022

Prof. Xingzhong Liu
Nankai University
College of Life Science
No.94 Weijin Road, Nankai District
Tianjin 300071
China

Re: Spectrum01872-22R1 (SNARE protein DdVam7 of nematode-trapping fungus *Drechlerella dactyloides* regulates vegetative growth, conidiation and predatory process via vacuole assembly)

Dear Prof. Xingzhong Liu:

Your manuscript has been accepted, and I am forwarding it to the ASM Journals Department for publication. You will be notified when your proofs are ready to be viewed.

Sincerely,

Chengshu Wang
Editor, Microbiology Spectrum

Journals Department
Supplemental file 4: Accept
Supplemental file 2: Accept
Supplemental file 3: Accept
Supplemental Material: Accept
Supplemental file 5: Accept
Supplemental file 1: Accept